# Gastric Myoelectric Activity and Body Composition Changes during Weight Loss via Sleeve Gastrectomy and Lifestyle Modification: Prospective Cohort Study

**DOI:** 10.3390/healthcare11081105

**Published:** 2023-04-12

**Authors:** Mahmoud M. A. Abulmeaty, Dara Aldisi, Mohamed Al Zaben, Ghadeer S. Aljuraiban, Abdulaziz Alkhathaami, Ali M. Almajwal, Eman El Shorbagy, Yara Almuhtadi, Zaid Aldossari, Thamer Alsager, Suhail Razak, Mohamed Berika

**Affiliations:** 1Community Health Sciences Department, College of Applied Medical Sciences, King Saud University, Riyadh 11433, Saudi Arabia; 2Obesity Management Unit, Medical Physiology Department, School of Medicine, Zagazig University, Zagazig 44519, Egypt; 3Surgery Department, Sultan Bin Abdulaziz Humanitarian City, Riyadh 13571, Saudi Arabia; 4Rehabilitation Sciences Department, College of Applied Medical Sciences, King Saud University, Riyadh 11433, Saudi Arabia

**Keywords:** gastric myoelectric activity, bariatric surgery, lifestyle intervention, weight loss

## Abstract

The capability of bariatric surgery (BS) and lifestyle intervention (LSI) in ameliorating obesity-associated altered gastric myoelectric activity (GMA) in relation to body composition is underinvestigated. This work studied GMA during weight loss via sleeve gastrectomy and multimodal lifestyle intervention. Seventy-nine participants with morbid obesity were assigned into three groups: bariatric surgery (BS group, *n* = 27), in which laparoscopic sleeve gastrectomy was performed; lifestyle intervention (LS group, *n* = 22), in which a calorie-deficit balanced diet with gradual physical activity and personalized behavioral modification were carried out; and waitlist control (C group, *n* = 30). For all participants, multichannel electrogastrography (EGG) with water-load testing and bioelectric impedance body composition analysis were done at baseline, after three months, and at six months. In the BS group, the water-load volume was decreased but without improvement in the bradygastria. In the LS group, preprandial bradygastria were reduced and some postprandial normogastria were increased throughout the study period. Except for fat-free mass and total body water, the parameters of body composition changes were superior in the BS group. In the LS group, the amount of fat-mass loss was negatively correlated with bradygastria times and positively correlated with preprandial and the early postprandial average dominant frequency (ADF). In addition, in the BS group, fat-mass loss was positively correlated with the ADF at late postprandial times. In conclusion, compared to BS, LS produced moderate normalization of GMA with the preservation of fat-free mass. The GMA changes were significantly associated with the amount of fat loss, regardless of the method of obesity management.

## 1. Introduction

The foundational step in obesity management is lifestyle intervention (LSI), including diet, increased physical activity, and behavioral modification. For patients who struggle with weight loss via LSI, adding medications that have been approved for chronic weight management is required [1]. For resistant cases with morbid obesity, surgical intervention in the form of sleeve gastrectomy, Roux-en-Y gastric bypass, and other procedures can produce sustained weight loss in addition to significant health improvement [2]. Despite the modest (5–10%) weight loss in LSI programs, many health benefits have been reported. Lifestyle modification programs do not usually produce marked weight changes, creating a mismatch between a patient’s goals and real achievement. However, with bariatric surgery, weight loss is much more common than the proven health benefits [3].

Obesity has been linked to several comorbidities and chronic diseases [4], including disturbances in the gastrointestinal tract (GIT), especially gastric functions. It has been reported that obesity is associated with functional GIT disorders, pancreatitis, inflammatory bowel disease, and GIT cancer [5]. Disturbed gastric myoelectrical activity (GMA) is among the obesity comorbidities. It has been reported that children with obesity have an increased state of impaired GMA, in comparison to other children [6]. In addition, adults with morbid obesity showed increased bradygastria times in both fasting and fed states [7]. Our previous reports stated that obesity produces distinctive patterns of GMA disturbance in different phenotypes of the disease, with an increased predominance of bradygastria rhythm [8].

Measuring GMA via multichannel transcutaneous electrogastrography (EGG) is a promising clinical procedure, especially with advanced technology devices that have received FDA approval. The hypothesis that “amelioration of obesity-induced GMA disturbances may be a mechanism of weight loss after BS” is a growing area of research. One study evaluated 20 patients undergoing laparoscopic non-adjustable gastric banding for 24 h, but GMA changes were insignificant [9]. Another study was conducted for three months after two different BS procedures (laparoscopic adjustable gastric banding and vertical banded gastroplasty); however, that study reported no clinically significant differences in GMA between the two procedures [10]. In addition, Dua et al. [11] studied GMA for a few days after pancreaticoduodenectomy to identify the best time to start oral feeding after surgery and to predict patients at risk for postoperative delayed gastric emptying. 

Despite the disappointing early reports, it is logical that BS can induce changes in GIT motility that are a direct effect of or secondary to an amelioration of previously obesity-induced disorders. Alterations of GMA and GIT motility are usually part of the mechanism of action and a result of BS [12]. The study of GMA for long periods after weight-loss surgeries or LSI programs is lacking in the literature and deserves greater investigation.

Body composition changes after BS versus LSI are frequently studied [13]. However, comprehensive studies of body composition in conjunction with measured resting energy expenditure and intensive dietary assessment in patients who have undergone BS compared to LSI programs are very limited in the literature. In this study, we investigated changes in GMA together with changes in body composition, energy expenditure, and dietary intake after sleeve gastrectomy, in contrast to changes after LSI, in a prospective manner for six months.

## 2. Materials and Methods

### 2.1. Study Subjects

Seventy-nine participants were divided into three groups: waitlist control (C group, *n* = 30), lifestyle intervention (LS group, *n* = 22), and bariatric surgery (BS group, *n* = 27) (Figure 1). The composition of the three groups was similar in terms of the BMI, age (18–40 years), and gender of the participants. Inclusion criteria included BMI > 35 kg/m^2^ with comorbidities or >40 kg/m^2^ without comorbidities and being free of any acute medical condition, malignancy, or psychiatric disorders. Persons who were receiving weight-losing pharmacotherapy or who had undergone previous bariatric surgery were excluded. The study outlines were explained to all participants before signing the informed written consent. The study protocol was reviewed and approved by the IRB committee of the College of Medicine, King Saud University, under reference number 20/0908/IRB, dated 30 November 2020, and by the research center at Sultan Bin Abdulaziz Humanitarian City (SBAHC) under reference number 49-2021-IRB, dated 3 June 2021. This study was also registered in ClinicalTrials.gov (record identifier is NCT05775172).

### 2.2. Interventions

For participants in the BS group, laparoscopic sleeve gastrectomy was performed under general anesthesia. The same surgery team from the SBAHC medical center managed all cases. The routine surgical preparation, procedures, and postoperative care (including a dietary regimen for BS patients) were carried out. Baseline measurements were taken during the preoperative preparation workout (a few days before the surgery date); then. every patient was given two post-operative appointments for the third month (third M assessment) and the sixth month (sixth M assessment), so that all study measurements could be carried out once again.

For participants in the LS group, a baseline assessment was performed at the first visit before the start of the intervention; then, every patient was provided with a schedule of biweekly short appointments for follow-up and two long appointments for the third- month and sixth-month assessments. Lifestyle intervention included a balanced 500- to 1000-calorie-deficit diet, a personalized physical activity plan, and customized behavioral modification. Detailed LS interventions were carried out according to our previous publication [14].

For the no-intervention control (C) group, three appointments were made for baseline, third M, and sixth M assessments without providing any weight loss instructions.

### 2.3. Gastric Myoelectric Activity

GMA was measured for 10 min after fasting for about 10–12 h and then again after water loading for 30 min by using a multichannel electrogastrography (EGG) with a water-load satiety test (3CPM-EGG, Sparks, MD, USA) [15]. EGG recording was carried out in a quiet room with low illumination while the patient was lying flat in an examination bed. The EGG disposable electrodes were applied to the skin at the midline between the xiphisternum and umbilicus (black electrode) and two inches below costal cartilage at the midclavicular line on the right side (green electrode) and left side (red electrode) (Figure 2) [16]. The EGG parameters that were used for analysis included the distribution of average power by frequency region as a percentage of bradygastria (1.0–2.5 cpm), normogastria (2.5–3.75 cpm), tachygastria (3.75–10.0 cpm), and duodenal respiration (10.0–15.0 cpm), as well as the average dominant frequency (ADF) in preprandial period (ADF-PR) and each 10 min of the postprandial recording period (ADF-min10, ADF-min20, and ADF-min30). The ADF is the dominant frequency, which is believed to be of gastric origin (not a noise), at which the power in the power spectrum has a peak value. The normal range of the ADF is between 2 to 4 cpm [17]. 

### 2.4. Anthropometric, Body Composition Measurement, and before and after Changes

Weight, height, and body mass index were measured as usually reported. A multi-frequency segmental bioelectric impedance analyzer (Tanita BC-418, Tanita Corp., Tokyo, Japan) was used to analyze body composition. The parameters used for analysis included percentage body fat (PBF), fat mass (FM), fat-free mass (FFM), the fat-mass index (FMI = FM/height^2^), the fat-free mass index (FFMI = FFM/height^2^) [18], the visceral fat (VF) rating, and total body water (TBW). The weight-loss amount was calculated in kg according to the following equation [Wt loss= baseline weight − 6-month weight]; the percentage of weight change was calculated according to [% of Wt change = (baseline weight − 6-month weight/baseline weight) × 100]; and percentage of excess-weight loss was calculated according to [% of excess Wt loss = {(baseline BMI − 6-month BMI)/(baseline BMI − 22.3)} × 100]. The amount of fat-mass loss was calculated by using the following equation [FM loss (kg) = baseline fat mass − 6-month fat mass] and the percentage of excess-body-fat loss was calculated based on the fat-mass index (FMI) as follows [(initial FMI − 6-month FMI)/(initial FMI − ideal FMI) × 100] [8]. The ideal FMI was indicated according to our population-specific cutoff (9.7 kg/m^2^ for women and 6.3 kg/m^2^ for men) [18].

### 2.5. Statistical Analysis

The Shapiro–Wilk test was used to test the normality of the study variables. The study parameters were compared among study groups by one-way ANOVA and Tukey’s HSD test for post hoc analysis or by a Kruskal–Wallis H test with the intergroup comparisons with the Mann–Whitney U test. Repeated measurements were assessed by Friedman’s ANOVA with Kendall’s coefficient of concordance for multiple comparisons of all pairwise. *p*-values < 0.05 were considered statistically significant. The Pearson correlation coefficient was used to test the correlation of EGG parameters at the six-month assessment with the amount of weight loss and fat-mass loss. SPSS software version 25 (SPSS Inc., Chicago, IL, USA) was used for all analyses.

## 3. Results

### 3.1. Baseline Characteristic Data among Study Groups

A comparison of the baseline assessment of the three groups is shown in Table 1. Age had no significant difference. Females represented 46% of the C group, 50% of the LS group, and 44% of the BS group. Body composition parameters were similar in the three groups (*p* > 0.5). Apart from a lower percentage of normogastria in the LS group, other parameters, including water-loading volume, were insignificantly different among the three groups.

### 3.2. Final Characteristic Data among Study Groups

At the date of the six-month assessment, some dropouts occurred—e.g., the C group had 30% no-shows, the LS group had 9% no-shows, and the BS no-show percent was 14.8%. The main causes of the dropouts were lack of interest (33%), especially in the C group, and living outside Riyadh (75%), especially in the BS group. The water-loading gastric volume was significantly lower in the BS group (Table 2). However, the other parameters of the EGG were insignificantly different among the groups. Regarding body composition changes, the BS group was superior to the LS in the reduction of BMI, the percentage of body fat, FM, FMI, and the VF rate. However, changes in FFM, the FFMI, muscle mass, and total body water were insignificantly different.

### 3.3. Third-Month and Six-Month Changes in the EGG

In the BS group, water-load volume reduced after three months with no additional significant reduction at the six-month assessment (Table 3), while both the C and LS groups showed insignificant changes in the water-load volumes (i.e., no reduction in the gastric volume). In the LS group, the percentage of preprandial bradygastria times was significantly reduced at the third-month assessment versus the baseline assessment (40.6 ± 15.7% vs. 65.6 ± 17.9%). Moreover, the percentage of normogastria increased during the Min30 period during the six-month assessment (19.5 ± 7.4 vs. 13.6 ± 5.5%).

The condition of the BS group was different—i.e., apart from an increment of the Min20 duodenal rhythm at the six-month assessment, other parameters did not change. In summary, BS reduced the gastric volume and slowed down the high duodenal rhythm pattern, while LS kept the gastric volume stable but reduced the bradygastria time and increased the normogastria time.

### 3.4. Third-Month and Sixth-Month Changes in the Body Composition

A comparison of the third-month and sixth-month assessments with the baseline body composition parameters is shown in Table 4. In the C group, despite the absence of lifestyle intervention, BMI was significantly reduced only at the sixth-month assessment, while other body-composition parameters were insignificantly changed throughout the study’s duration. In the LS group, there were reductions in the BMI, PBF, FM, FMI, FFM, FFMI, muscle mass, VF rate, and TBW after three months, with no additional significant reductions in the sixth month. In the BS group, there were progressive reductions in BMI, FM, and FMI with time. No further significant reductions in the PBF, the VF rate, FFM, FFMI, and muscle mass after the middle assessment. Moreover, TBW was insignificantly different in the three-time points. At the sixth-month assessment, the weight loss, the percentage of weight change, the percentage of excess weight loss, the amount of fat-mass loss, and the percentage of excess body fat loss were significantly higher in the LS group than in the C group and higher in the BS group than in the LS group, as shown in Figure 3.

### 3.5. Correlation of Sixth-Month Weight Loss and Fat-Mass Loss with EGG Parameter

As shown in Table 5, the FM loss in the C group showed a significant inverse correlation with the percentages of normogastria at the first 10 min of postprandial recording and a positive correlation with the percentage of the bradygastria during the same period. Notably, the FM loss values in the C group were negative (i.e., gaining fat), leading to the conclusion that with increasing gains in fat mass, the percentage of normogastria reduced and the percentage of bradygastria increased.

In the LS group, the amount of fat-mass loss was inversely correlated with the percentage of preprandial bradygastria. In addition, FM loss showed a significant correlation with ADF at preprandial recording and the middle 10 min of postprandial recording (r = 0.645, 0.556, respectively, *p* < 0.05). Furthermore, the amount of weight loss showed a positive correlation with the percentages of duodenal rhythm and tachygastria times, as well as a negative correlation with the percentage of bradygastria times (Table 5).

In the BS group, FM loss correlated positively with the ADF during most of the postprandial recording period and with the percentage of duodenal rhythm. Moreover, weight loss showed a positive correlation with the ADF at 30 min.

## 4. Discussion

This study compared GMA and body composition before and after BS or LSI in a prospective manner. The most interesting finding is that sleeve gastrectomy reduced the stomach’s volume (denoted by the reduction in water-load volume) without significantly improving the bradygastria status (there was only an increase in the percentage of duodenal rhythm). While LSI failed to reduce gastric volumes, LSI produced some significant improvement in bradygastria and normogastria times. Regarding BS, the previous reports of Gürlich et al. [9], van Dielen et al. [10], and Crittenden et al. [19] identified insignificant changes in GMA after short-term periods after BS (24 h, three months, and three months, respectively), and they studied other BS procedures (gastric banding and vertical banded gastroplasty). 

Crittenden et al. [19] used EGG together with other measures of the autonomic nervous system (ANS) for predicting long-term weight regain 15 years after vertical-banded gastroplasty. They found an insignificant difference between the weight-gainer group and the weight-loser group regarding EGG, which was measured preoperatively and three months after surgery. In addition, they reported that a multi-component model, including EGG and ANS parameters at baseline and three months postoperatively, could predict long-term weight outcomes after 15 years. In the current study, we followed the patients for six months and studied another procedure—sleeve gastrectomy—in which a huge part of the greater curvature was removed, including parts of the fundus and the body that are the anatomical sites of the interstitial cell of Cajal (ICC) based on the proto-oncogene c-Kit staining. The ICC is the origin of slow-wave propagations that are the targets for recording by EGG [20]. This may explain the under-expected effect of sleeve gastrectomy on GMA. Loss of ICC was suggested as a mechanism for gastroparesis and dumping [21,22]. 

Contrary to our findings, Robertson et al. [23] compared EGG findings in sleeve gastrectomy patients (at only three months after surgery) with matched healthy controls (i.e., a case-control pattern) and found that gastric rhythm was significantly lower in patients compared to controls and GMA amplitude was also lower in sleeve gastrectomy patients. This difference between our results and those of Robertson et al. resulted from using different devices and different study designs. In our study, we used a clinically valid FDA-approved machine and compared sleeve gastrectomy patients with BMI-matched controls in a prospective manner. Correlations were also supportive of the hypothesis that weight/fat-mass loss increased the ADF of the stomach. This effect of weight/fat mass loss was present in both the LS and BS groups.

In other disease populations, Kim et al. [24] used EGG in the assessment of GMA in gastric remnants after distal gastrectomy for stomach cancer and found that the tachygastria percentage was increased, whereas the bradygastria percentage was decreased during the postoperative periods. The effects of other surgical procedures on EGG were also studied with interesting results. Gastrostomy tube (PEG) insertion in neurologically impaired children does not lead to abnormal EGG tracing, while the anti-reflux fundoplication procedure leads to marked dysrhythmic EGG [25]. Patients with gastroesophageal reflux (GERD) also have fewer percentages of normogastria [26]. Preprandial and postprandial GMA after cholecystectomy, subtotal gastrectomy, or vagotomy, and Nissen fundoplication in adults, showed greater dysrhythmias than those of normal volunteers [27]. In cases with ischemic gastroparesis, the abnormal GMA rhythm resolves after vascular repair [26].

To the best of our knowledge, this is the first work studying the effect of multimodal LSI, including diet, physical activity, and behavioral modification, on EGG prospectively. However, some previous reports investigated the effect of physical activity on only EGG (usually in normal participants). Lu et al. [28] found that the use of a cycle ergometer for 10 min with 50% of the maximum age-predicted heart rate induced a more regular and more stable gastric slow wave of higher amplitude. However, another study found that high-intensity exercise slows down gastric rhythm [29]. In general, there is limited evidence suggesting that chronic exercise is associated with faster gastric emptying [30]. A study of gastric emptying (GE) as an important consequent function of GMA was previously carried out after weight-loss intervention by non-surgical methods. Verdich et al. [31] found that weight loss was associated with a reduction in GE, measured by 99 m Tc-labeled sulfur colloid tracing, only during the initial 30 min after a test meal.

This study showed greater improvement in body composition parameters in the sleeve gastrectomy group than in the LS group. A recent meta-analysis concluded that sleeve gastrectomy leads to changes in weight loss, fat-mass loss, and lean-mass loss that are similar to the changes following a Roux-en-Y gastric bypass (RYGB). Moreover, RYGB led to a greater total-weight loss, greater fat-mass loss, and similar lean-mass loss than gastric banding [32]. Another systematic meta-analysis compared BS with LSI and concluded that BS produced greater weight loss, higher remission rates of metabolic syndrome and type 2 diabetes, and better metabolic laboratory parameters [33]. Interestingly, the current study revealed that fat-mass change is associated with GMA changes in both the LS and BS groups. The means of the ADF-BL and the ADF-min20 were positively correlated with fat-mass loss in the LS group (r = 0.645 and 0.556, respectively, *p* < 0.05). In the BS group, the means of the ADF-20 and the ADF-min30 were positively correlated with fat-mass loss (r = 0.448 and 0.774, respectively, *p* < 0.05). This may postulate a mechanism for GMA changes after weight loss, regardless of the method. The normalization of GMA by surgery or successful LS intervention may work peripherally to reduce the amount of food intake and modulate gastric emptying, as well as centrally via gastric-brain coupling to affect satiety. Rebollo et al. [34] found that the functional MRI studies showed that the gastric rhythm is coupled to the brain in many cortical pathways, including primary and secondary somatosensory cortices and in the parieto-occipital region. The ingestion of food stimulates mechanosensitive and stretch receptors in the stomach, sending the vagal sensory drive to cortical and subcortical brain areas to produce the satiation perception [35].

Despite some of the strengths of this article, we encountered some limitations. The main limitation was the uneven dropout rate. High dropout rates are frequent in prospective studies of obesity management over a long period [36]. Another limitation was using only the EGG for assessment of GMA, without combined gastric-emptying studies using scintigraph.

## 5. Conclusions

In the BS group, gastric volume was decreased but without improvement in the state of bradygastria. In the LS group, preprandial bradygastria percentages were reduced and some postprandial normogastria percentages were increased throughout the study period. Fat mass and percent of excess fat losses were superior in the BS group. Despite the superior weight-losing effect of BS, LS produced moderate normalization of GMA with the preservation of fat-free mass. GMA changes were significantly associated with the amount of fat loss, regardless of the method of obesity management.

## Figures and Tables

**Figure 1 healthcare-11-01105-f001:**
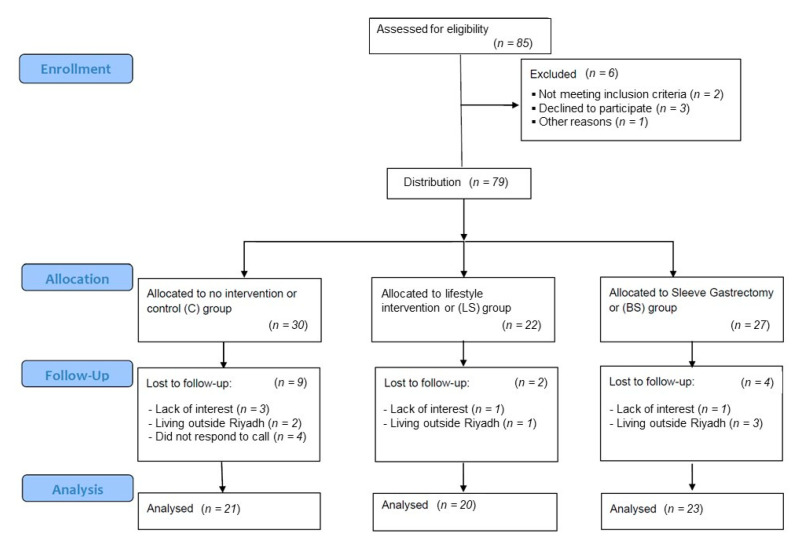
Flow diagram of the progress through the phases of the study.

**Figure 2 healthcare-11-01105-f002:**
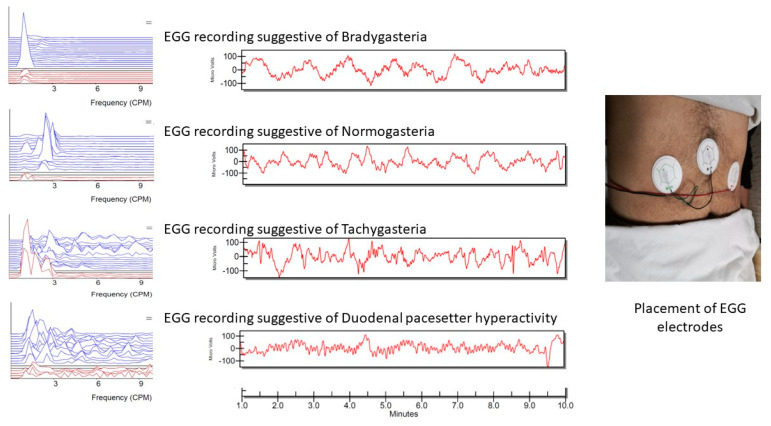
Placement or EGG electrodes and examples of EGG patterns. Bradygastria is defined as a decreased rate of myoelectrical activity in the stomach <2.5 cycles per minute (cpm). Normogastria is a slow wave frequency at 2.5 to 3.75 cpm; tachygastria is a frequency from 3.75 to 10 cpm; the duodenal pacesetter hyperactivity indicates a higher frequency of 10 to 15 cpm.

**Figure 3 healthcare-11-01105-f003:**
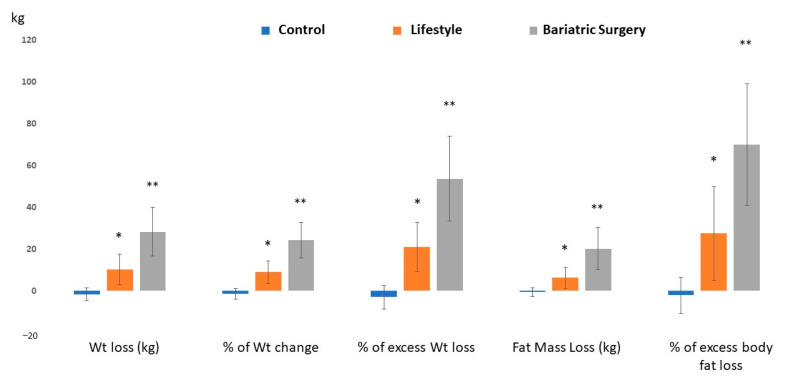
Weight and fat mass changes before and after intervention in all study groups. * Significant versus the C group; ** significant versus both C and LS groups.

**Table 1 healthcare-11-01105-t001:** Baseline data of all study groups.

Variables	C (*n* = 30) Mean ± SD	LS (*n* = 22) Mean ± SD	BS (*n* = 27) Mean ±SD	*p*-Value
Age	34.87 ± 13.14 ^a^	37.36 ± 13.36 ^a^	32.11 ± 10.72 ^a^	0.529
Electrogastrography				
Water load (mL)	551.33 ± 263.22 ^a^	716.82 ± 298.39 ^a^	607.75 ± 271.22 ^a^	0.290
PR-BradayG (%)	57.08 ± 22.38 ^a^	57.30 ± 28.54 ^a^	47.89 ± 18.14 ^a^	0.413
PR-NromoG (%)	14.12 ± 12.49 ^a^	11.28 ± 7.42 ^a^	15.25 ± 4.71 ^a^	0.494
PR-TachyG (%)	18.41 ± 9.51 ^a^	23.83 ± 24.35 ^a^	27.77 ± 13.43 ^a^	0.250
PR-Duodenal (%)	10.39 ± 10.43 ^a^	7.59 ± 10.85 ^a^	9.09 ± 7.04 ^a^	0.750
Min10_BradyG (%)	59.02 ± 17.69 ^a^	63.08 ± 26.88 ^a^	62.64 ± 16.76 ^a^	0.837
Min10_NormoG (%)	14.83 ± 8.80 ^a^	7.38 ± 4.11 ^b^	12.69 ± 6.67 ^a^	0.033 0
Min10_TachyG (%)	20.92 ± 12.55 ^a^	19.42 ± 17.92 ^a^	18.27 ± 10.03 ^a^	0.847
Min10_Duodenal (%)	5.22 ± 5.16 ^a^	10.13 ± 16.67 ^a^	6.41 ± 8.26 ^a^	0.471
Min20_BradyG (%)	51.69 ± 12.93 ^a^	56.38 ± 28.22 ^a^	56.91 ± 17.80 ^a^	0.722
Min20_NormoG (%)	16.43 ± 6.55 ^a^	11.43 ± 4.51 ^a^	17.26 ± 10.14 ^a^	0.146
Min20_TachyG (%)	25.16 ± 8.06 ^a^	21.44 ± 20.52 ^a^	18.93 ± 10 ^a^	0.393
Min20_Duodenal (%)	6.72 ± 5.14 ^a^	10.75 ± 17.17 ^a^	6.89 ± 7.91 ^a^	0.553
Min30_BradyG (%)	51.65 ± 9.64 ^a^	51.16 ± 21.23 ^a^	54.88 ± 16.83 ^a^	0.784
Min30_NormoG (%)	21.02 ± 9.49 ^a^	14.92 ± 6.89 ^a^	17.08 ± 7.80 ^a^	0.163
Min30_TachyG (%)	22.41 ± 7.79 ^a^	24.73 ± 16.29 ^a^	21.79 ± 10.52 ^a^	0.792
Min30_Duodenal (%)	4.92 ± 2.80 ^a^	9.20 ± 7.79 ^a^	6.26 ± 8.12 ^a^	0.279
ADF-PR (cpm)	1.66 ± 0.88 ^a^	2.34 ± 2.45 ^a^	1.57 ± 0.94 ^a^	0.465
ADF-min10 (cpm)	1.48 ± 0.38 ^a^	2.31 ± 2.32 ^a^	1.58 ± 0.41 ^a^	0.185
ADF-min20 (cpm)	1.59 ± 0.66 ^a^	3.08 ± 3.98 ^a^	1.58 ± 0.84 ^a^	0.132
ADF-min30 (cpm)	1.19 ± 0.46 ^a^	1.68 ± 1.67 ^a^	1.46 ± 1.42 ^a^	0.659
Body composition				
BMI (kg/m^2^)	40.70 ± 4.01 ^a^	39.86 ± 2.91 ^a^	43.30 ± 6.35 ^a^	0.147
Fat (%)	44.13 ± 5.90 ^a^	41.65 ± 7.02 ^a^	44.59 ± 6.30 ^a^	0.464
Fat mass (kg)	49.64 ± 7.09 ^a^	45.91 ± 8.54 ^a^	51.81 ± 12.10 ^a^	0.299
Fat mass index (kg/m^2^)	17.94 ± 2.88 ^a^	16.65 ± 3.27 ^a^	19.12 ± 4.26 ^a^	0.211
Visceral fat rate	15.13 ± 4.98 ^a^	16.73 ± 4.27 ^a^	17.56 ± 5.62 ^a^	0.401
Fat-free mass (kg)	63.97 ± 14.42 ^a^	65.01 ± 13.28 ^a^	64.24 ± 12.22 ^a^	0.980
Fat-free mass index (kg/m^2^)	22.76 ± 3.39 ^a^	23.21 ± 2.94 ^a^	23.43 ± 2.68 ^a^	0.813
Muscle mass (kg)	60.87 ± 13.82 ^a^ (7.7)	61.88 ± 12.79 ^a^	61.11 ± 11.71 ^a^	0.979
Total body water (kg)	46.83 ± 10.54 ^a^	47.58 ± 9.71 ^a^	46.96 ± 8.88 ^a^	0.979

C = control group; LS = lifestyle intervention group; BS = bariatric surgery group; different superscripts (a,b) indicate statistically different. BradyG = bradygasria; PR = preprandial; NormoG = normogastria; TachyG = tachygastria. Min10 = the first 10 min in postprandial recording; Min20 = the period between 10 to 20 min in postprandial recording; Min30 = the period between 20 to 30 min in postprandial recording; ADF = average dominant frequency. BMI = body mass index.

**Table 2 healthcare-11-01105-t002:** Six-month assessment in the three study groups.

Variables	C (*n* = 21) Mean ± SD	LS (*n* = 20) Mean ± SD	BS (*n* = 23) Mean ± SD	*p*-Value
Electrogastrography				
Water load (mL)	457.17 ± 161.73 ^a^	631.29 ± 209.03 ^a^	328.00 ± 207.26 ^b^	0.050
PR-BradayG (%)	48.55 ± 22.44 ^a^	47.32 ± 24.54 ^a^	58.13 ± 27.76 ^a^	0.736
PR-NromoG (%)	17.01 ± 8.00 ^a^	9.51 ± 3.51 ^a^	9.17 ± 4.38 ^a^	0.049
PR-TachyG (%)	26.00 ± 16.54 ^a^	29.97 ± 15.79 ^a^	26.18 ± 21.67 ^a^	0.903
PR-Duodenal (%)	8.43 ± 9.76 ^a^	13.19 ± 12.27 ^a^	6.62 ± 4.19 ^a^	0.495
Min10_BradyG (%)	64.91 ± 12.94 ^a^	60.60 ± 14.62 ^a^	60.68 ± 14.24 ^a^	0.831
Min10_NormoG (%)	11.02 ± 2.72 ^a^	12.38 ± 5.46 ^b^	11.02 ± 6.90 ^a^	0.887 0
Min10_TachyG (%)	18.79 ± 7.52 ^a^	21.64 ± 10.14 ^a^	19.40 ± 5.91 ^a^	0.812
Min10_Duodenal (%)	5.12 ± 4.30 ^a^	5.38 ± 4.26 ^a^	8.57 ± 11.95 ^a^	0.687
Min20_BradyG (%)	54.23 ± 24.49 ^a^	52.13 ± 17.57 ^a^	46.63 ± 13.63 ^a^	0.804
Min20_NormoG (%)	12.24 ± 5.20 ^a^	14.65 ± 4.68 ^a^	9.10 ± 4.94 ^a^	0.191
Min20_TachyG (%)	25.15 ± 15.21 ^a^	27.49 ± 11.82 ^a^	23.81 ± 9.71 ^a^	0.876
Min20_Duodenal (%)	8.37 ± 6.36 ^a^	5.73 ± 7.04 ^a^	20.46 ± 15.62 ^a^	0.058
Min30_BradyG (%)	54.70 ± 15.58 ^a^	51.07 ± 14.03 ^a^	51.89 ± 17.28 ^a^	0.911
Min30_NormoG (%)	14.70 ± 9.3 ^a^	19.46 ± 7.43 ^a^	10.14 ± 3.23 ^a^	0.127
Min30_TachyG (%)	222.62 ± 8.01 ^a^	24.22 ± 10.80 ^a^	20.24 ± 7.30 ^a^	0.760
Min30_Duodenal (%)	7.98 ± 6.24 ^a^	5.26 ± 3.68 ^a^	17.71 ± 15.71 ^a^	0.096
ADF-PR (cpm)	3.45 ± 2.78 ^a^	1.90 ± 1.02 ^a^	1.63 ± 1.28 ^a^	0.228
ADF-min10 (cpm)	1.48 ± 0.38 ^a^	2.31 ± 2.32 ^a^	1.58 ± 0.41 ^a^	0.836
ADF-min20 (cpm)	1.46 ± 0.37 ^a^	1.40 ± 0.24 ^a^	1.60 ± 0.97 ^a^	0.561
ADF-min30 (cpm)	1.44 ± 0.53 ^a^	2.01 ± 0.83 ^a^	2.03 ± 1.66 ^a^	0.318
Body composition				
BMI (kg/m^2^)	38.37 ± 3.89 ^a^	35.71 ± 3.60 ^a^	26.66 ± 2.50 ^b^	0.016
Fat (%)	47.90 ± 5.00 ^a^	39.30 ± 9.22 ^a^	27.45 ± 6.9 ^b^	0.040
Fat mass (kg)	52.57 ± 12.08 ^a^	37.77 ± 10.32 ^a^	20.85 ± 4.78 ^b^	0.013
Fat mass index (kg/m^2^)	19.42 ± 4.16 ^a^	14.23 ± 4.21 ^a^	7.37 ± 1.97 ^b^	0.023
Visceral fat rate	16.33 ± 6.66 ^a^	12.50 ± 5.05 ^a^	7.75 ± 4.35 ^a^	0.144
Fat-free mass (kg)	54.77 ± 14.09 ^a^	58.57 ± 13.35 ^a^	55.70 ± 10.37 ^a^	0.775
Fat-free mass index (kg/m^2^)	20.05 ± 3.32 ^a^	21.47 ± 2.19 ^a^	19.30 ± 1.97 ^a^	0.431
Muscle mass (kg)	52.10 ± 13.55 ^a^ (7.7)	55.81 ± 12.82 ^a^	53.13 ± 9.98 ^a^	0.775
Total body water (kg)	40.10 ± 10.31 ^a^	42.88 ± 9.77 ^a^	45.25 ± 4.17 ^a^	0.794

C = control group; LS = lifestyle intervention group; BS = bariatric surgery group; different superscripts (a,b) indicate statistically different. BradyG = bradygasria; PR = preprandial; NormoG = normogastria; TachyG = tachygastria. Min10 = the first 10 min in postprandial recording; Min20 = the period between 10 to 20 min in postprandial recording; Min30 = the period between 20 to 30 min in postprandial recording; ADF = average dominant frequency. BMI = body mass index.

**Table 3 healthcare-11-01105-t003:** Time-related changes in the EGG parameters among study groups.

Variables	C (*n* = 21)				LS (*n* = 20)				BS (*n* = 23)			
	Baseline	3-Month	6-Month	*p*-Value	Baseline	3-Month	6-Month	*p*-Value	Baseline	3-Month	6-Month	*p*-Value
Water load (mL)	660.0 ± 361.5 ^a^	531.7 ± 264.2 ^a^	457.2 ± 161.7 ^a^	0.257	642.1 ± 232.2 ^a^	583.7 ± 234.6 ^a^	631.3 ± 209.0 ^a^	0.368	630.3 ± 117.4 ^a^	317.1 ± 198.0 ^b^	328.3 ± 207.2 ^b^	0.019
PR-BradayG (%)	67.1 ± 16.4 ^a^	60.4 ± 23.5 ^a^	48.6 ± 22.4 ^a^	0.438	65.6 ± 17.9 ^a^	40.6 ± 15.7 ^b^	47.3 ± 24.4 ^a^	0.021	60.0 ± 19.3 ^a^	51.1 ± 29.6 ^a^	58.1 ± 27.8 ^a^	0.949
PR -NromoG (%)	11.6 ± 5.3 ^a^	14.4 ± 6.1 ^a^	17.0 ± 8.0 ^a^	0.084	14.1 ± 7.8 ^a^	19.8 ± 13.7 ^a^	9.5 ± 3.5 ^a^	0.341	16.8 ± 7.1 ^a^	8.9 ± 3.6 ^a^	9.2 ± 4.4 ^a^	0.076
PR-TachyG (%)	15.0 ± 7.2 ^a^	19.3 ± 15.2 ^a^	26.0 ± 16.5 ^a^	0.438	16.2 ± 9.5 ^a^	26.4 ± 10.2 ^a^	30.0 ± 15.8 ^a^	0.054	20.1 ± 15.4 ^a^	21.9 ± 12.7 ^a^	26.2 ± 21.7 ^a^	0.854
PR-Duodenal (%)	6.3 ± 7.2 ^a^	5.8 ± 4.9 ^a^	8.4 ± 9.8 ^a^	0.738	4.0 ± 2.8 ^a^	13.2 ± 13.3 ^a^	13.2 ± 12.3 ^a^	0.857	3.1 ± 1.7 ^a^	18.1 ± 16.9 ^a^	6.6 ± 4.2 ^a^	0.854
Min10_BradyG (%)	62.5 ± 20.4 ^a^	75.4 ± 10.9 ^a^	64.9 ± 12.9 ^a^	0.200	75.1 ± 11.5 ^a^	53.3 ± 20.2 ^a^	60.6 ± 14.6 ^a^	0.054	68.9 ± 8.7 ^a^	59.7 ± 17.0 ^a^	60.7 ± 14.2 ^a^	0.854
Min10_NormoG (%)	14.2 ± 8.7 ^a^	10.5 ± 5.8 ^a^	11.01 ± 2.7 ^a^	0.957	8.2 ± 4.4 ^a^	14.7 ± 6.6 ^a^	12.4 ± 5.5 ^a^	0.341	15.2 ± 9.2 ^a^	12.0 ± 4.0 ^a^	11.4 ± 6.9 ^a^	0.949
Min10_TachyG (%)	18.9 ± 15.4 ^a^	11.5 ± 5.2 ^a^	18.8 ± 7.5 ^a^	0.200	12.8 ± 6.3 ^a^	26.5 ± 12.1 ^a^	21.6 ± 10.14 ^a^	0.085	13.7 ± 4.6 ^a^	23.5 ± 16.0 ^a^	19.4 ± 5.9 ^a^	0.241
Min10_Duodenal (%)	4.4 ± 4.2 ^a^	2.6 ± 1.9 ^a^	5.1 ± 4.3 ^a^	0.200	3.9 ± 3.5 ^a^	5.5 ± 5.6 ^a^	5.4 ± 4.3 ^a^	1.000	2.3 ± 0.7 ^a^	4.9 ± 3.3 ^a^	8.6 ± 11.9 ^a^	0.692
Min20_BradyG (%)	61.6 ± 11.6 ^a^	67.8 ± 10.4 ^a^	54.2 ± 24.5 ^a^	0.738	68.1 ± 14.5 ^a^	51.8 ± 18.9 ^a^	52.1 ± 17.6 ^a^	0.250	62.4 ± 7.3 ^a^	51.9 ± 28.5 ^a^	46.6 ± 13.6 ^a^	0.331
Min20_NormoG (%)	14.2 ± 5.6 ^a^	13.4 ± 6.9 ^a^	12.2 ± 5.2 ^a^	0.309	13.2 ± 3.9 ^a^	16.4 ± 5.9 ^a^	14.6 ± 4.6 ^a^	0.540	18.1 ± 7.6 ^a^	11.9 ± 7.5 ^a^	9.1 ± 4.9 ^a^	0.196
Min20_TachyG (%)	19.8 ± 5.8 ^a^	15.8 ± 6.5 ^a^	25.2 ± 15.2 ^a^	0.401	13.9 ± 6.9 ^a^	25.8 ± 11.3 ^a^	27.5 ± 11.8 ^a^	0.341	16.1 ± 3.1 ^a^	27.2 ± 21.8 ^a^	23.8 ± 9.7 ^a^	0.331
Min20_Duodenal (%)	4.3 ± 3.0 ^a^	2.9 ± 1.0 ^a^	8.4 ± 6.4 ^a^	0.337	4.8 ± 6.8 ^a^	5.9 ± 8.8 ^a^	5.7 ± 7.1 ^a^	0.857	3.3 ± 2.9 ^a^	9.0 ± 8.5 ^a^	20.5 ± 15.6 ^b^	0.021
Min30_BradyG (%)	53.1 ± 11.0 ^a^	54.8 ± 20.4 ^a^	54.7 ± 15.6 ^a^	0.738	56.6 ± 16.8 ^a^	59.4 ± 26.3 ^a^	51.1 ± 14.1 ^a^	0.630	49.2 ± 19.7 ^a^	47.4 ± 16.2 ^a^	51.9 ± 17.3 ^a^	0.504
Min30_NormoG (%)	21.1 ± 4.1 ^a^	21.2 ± 10.1 ^a^	14.7 ± 9.3 ^a^	0.957	13.6 ± 5.5 ^a^	10.9 ± 5.5 ^a^	19.5 ± 7.4 ^b^	0.013	20.1 ± 8.8 ^a^	10.4 ± 3.0 ^a^	10.1 ± 3.2 ^a^	0.076
Min30_TachyG (%)	22.2 ± 9.0 ^a^	18.0 ± 7.1 ^a^	22.6 ± 8.0 ^a^	0.200	22.1 ± 9.7 ^a^	23.6 ± 18.2 ^a^	24.2 ± 10.8 ^a^	0.630	20.6 ± 5.0 ^a^	24.5 ± 8.3 ^a^	20.2 ± 7.3 ^a^	0.949
Min30_Duodenal (%)	3.6 ± 1.7 ^a^	5.9 ± 5.8 ^a^	8.0 ± 6.2 ^a^	0.738	7.8 ± 7.3 ^a^	6.1 ± 7.1 ^a^	5.2 ± 3.7 ^a^	0.341	10.1 ± 14.9 ^a^	17.6 ± 8.3 ^a^	17.7 ± 15.9 ^a^	0.504
ADF-PR (cpm)	1.5 ± 0.7 ^a^	1.3 ± 0.2 ^a^	1.8 ± 0.8 ^a^	0.085	1.2 ± 0.3 ^a^	1.8 ± 0.4 ^a^	1.9 ± 1.0 ^a^	0.250	1.3 ± 0.6 ^a^	2.3 ± 1.5 ^a^	1.6 ± 1.3 ^a^	0.692
ADF-min10 (cpm)	1.3 ± 0.5 ^a^	1.1 ± 0.1 ^a^	1.5 ± 0.4 ^a^	0.257	1.3 ± 0.4 ^a^	1.7 ± 1.1 ^a^	1.4 ± 0.3 ^a^	0.887	1.5 ± 0.5 ^a^	1.6 ± 1.3 ^a^	1.6 ± 0.9 ^a^	0.801
ADF-min20 (cpm)	1.6 ± 0.2 ^a^	1.1 ± 0.3 ^a^	1.79 ± 0.4 ^a^	0.071	1.4 ± 0.3 ^a^	2.1 ± 1.5 ^a^	2.0 ± 0.8 ^a^	0.630	1.4 ± 0.3 ^a^	1.4 ± 0.2 ^a^	2.0 ± 1.7 ^a^	0.949
ADF-min30 (cpm)	1.4 ± 0.3 ^a^	1.0 ± 0.2 ^a^	0.6 ± 0.7 ^a^	0.368	1.1 ± 0.6 ^a^	1.4 ± 1.2 ^a^	1.2 ± 0.6 ^a^	0.857	2.5 ± 2.3 ^a^	1.1 ± 0.6 ^a^	2.8 ± 4.6 ^a^	0.196

C = control group; LS = lifestyle intervention group; BS = bariatric surgery group; different superscripts (a,b) indicate statistically different. BradyG = bradygasria; PR = preprandial; NormoG = normogastria; TachyG = tachygastria. Min10 = the first 10 min in postprandial recording; Min20 = the period between 10 to 20 min in postprandial recording; Min30 = the period between 20 to 30 min in postprandial recording; ADF = average dominant frequency.

**Table 4 healthcare-11-01105-t004:** Time-related changes in body composition among study groups.

Variables	C (*n* = 21)				LS (*n* = 20)				BS (*n* = 23)			
	Baseline	3-Month	6-Month	*p*-Value	Baseline	3-Month	6-Month	*p*-Value	Baseline	3-Month	6-Month	*p*-Value
BMI (kg/m^2^)	41.6 ± 5.4 ^a^	40.6 ± 5.6 ^a^	38.4 ± 3.9 ^b^	0.050	40.0 ± 3.5 ^a^	36.9 ± 2.8 ^b^	35.7 ± 3.6 ^b^	0.006	40.4 ± 4.2 ^a^	31.4 ± 2.5 ^b^	26.7 ± 2.5 ^c^	0.001
Fat (%)	48.3 ± 6.2 ^a^	47.4 ± 6.4 ^a^	47.9 ± 5.0 ^a^	0.761	42.3 ± 6.6 ^a^	40.2 ± 7.1 ^b^	39.3 ± 9.2 ^b^	0.012	45.0 ± 5.7 ^a^	33.3 ± 4.5 ^b^	27.5 ± 6.1 ^b^	0.001
Fat mass (kg)	51.9 ± 7.6 ^a^	49.7 ± 8.8 ^a^	52.6 ± 12.1 ^a^	0.368	45.5 ± 8.9 ^a^	39.8 ± 8.2 ^b^	37.8 ± 10.3 ^b^	0.006	52.6 ± 13.5 ^a^	29.9 ± 5.4 ^b^	20.9 ± 4.8 ^c^	0.001
Fat mass index (kg/m^2^)	20.0 ± 2.1 ^a^	19.1 ± 2.6 ^a^	19.4 ± 4.2 ^a^	0.368	17.0 ± 3.1 ^a^	14.9 ± 3.1 ^b^	14.2 ± 4.2 ^b^	0.006	18.2 ± 3.7 ^a^	10.5 ± 1.7 ^b^	7.4 ± 2.0 ^c^	0.001
Visceral fat rate	17.0 ± 5.2 ^a^	16.7 ± 5.5 ^a^	16.3 ± 6.7 ^a^	0.670	15.7 ± 5.1 ^a^	13.7 ± 4.8 ^b^	12.5 ± 5.0 ^b^	0.004	17.8 ± 6.2 ^a^	11.5 ± 5.4 ^b^	7.8 ± 4.3 ^b^	0.001
Fat-free mass (kg)	56.3 ± 14.5 ^a^	55.8 ± 6.1 ^a^	54.8 ± 14.1 ^a^	0.097	63.2 ± 16.4 ^a^	60.1 ± 14.7 ^b^	58.6 ± 13.3 ^b^	0.006	64.1 ± 13.1 ^a^	60.6 ± 11.9 ^b^	55.7 ± 10.4 ^b^	0.003
Fat-free mass index (kg/m^2^)	21.6 ± 5.1 ^a^	21.5 ± 5.1 ^a^	20.1 ± 3.3 ^a^	0.097	23.1 ± 3.0 ^a^	21.9 ± 2.6 ^b^	21.5 ± 2.2 ^b^	0.006	22.1 ± 2.3 ^a^	21.0 ± 2.1 ^b^	19.3 ± 2.0 ^b^	0.003
Muscle mass (kg)	53.6 ± 13.9 ^a^	53.1 ± 13.8 ^a^	52.1 ± 13.6 ^a^	0.097	60.1 ± 15.7 ^a^	57.3 ± 14.1 ^b^	55.8 ± 12.8 ^b^	0.006	61.0 ± 12.5 ^a^	57.8 ± 11.6 ^b^	53.1 ± 10.0 ^b^	0.003
Total body water (kg)	41.2 ± 10.6 ^a^	40.8 ± 10.5 ^a^	40.1 ± 10.3 ^a^	0.097	46.3 ± 12.0 ^a^	44.0 ± 10.8 ^b^	42.9 ± 9.8 ^b^	0.006	46.9 ± 9.6 ^a^	44.4 ± 8.8 ^a^	45.3 ± 4.2 ^a^	0.368

C = control group; LS = lifestyle intervention group; BS = bariatric surgery group; different superscripts (a–c) indicate statistically different. BMI = body mass index.

**Table 5 healthcare-11-01105-t005:** Correlation of weight loss and fat-mass loss with EGG parameters among study groups.

EGG Variables	C (*n* = 21)		LS (*n* = 20)		BS (*n* = 23)	
	Weight Loss (Kg)	Fat Mass Loss (Kg)	Weight Loss (Kg)	Fat Mass Loss (Kg)	Weight Loss (Kg)	Fat Mass Loss (Kg)
Water load (mL)	−0.363	−0.318	−0.038	0.090	0.226	0.046
PR-BradayG (%)	0.286	0.380	−0.311	−0.566 *	−0.246	0.054
PR -NromoG (%)	0.027	0.075	−0.418	−0.333	−0.195	−0.214
PR-TachyG (%)	−0.305	−0.416	−0.097	−0.380	0.252	−0.019
PR-Duodenal (%)	−0.245	−0.268	0.168	−0.082	0.233	0.017
Min10_BradyG (%)	0.437	0.669 *	0.300	0.417	0.094	−0.036
Min10_NormoG (%)	−0.232	−0.628 *	−0.419	−0.409	−0.012	−0.110
Min10_TachyG (%)	−0.388	−0.428	−0.360	−0.429	−0.196	0.182
Min10_Duodenal (%)	−0.277	−0.436	0.403	0.157	0.005	0.225
Min20_BradyG (%)	0.247	−0.014	−0.316	−0.266	0.174	−0.011
Min20_NormoG (%)	−0.420	0.002	−0.246	−0.166	−0.368	−0.381
Min20_TachyG (%)	−0.101	−0.074	0.261	0.292	−0.342	−0.316
Min20_Duodenal (%)	0.025	0.197	0.574 *	0.329	0.379	0.681 **
Min30_BradyG (%)	−0.074	0.144	−0.490 *	−0.392	−0.058	−0.132
Min30_NormoG (%)	0.421	0.073	0.230	0.206	−0.384	−0.449
Min30_TachyG (%)	−0.229	−0.265	0.570 *	0.441	−0.061	−0.223
Min30_Duodenal (%)	0.094	−0.050	0.049	0.036	0.229	0.408
ADF- PR (cpm)	−0.221	−0.247	0.333	0.645 *	0.065	0.046
ADF-min10 (cpm)	−0.291	−0.303	0.099	−0.140	0.065	0.308
ADF-min20 (cpm)	−0.387	−0.217	0.445	0.556 *	0.190	0.488 *
ADF-min30 (cpm)	−0.376	−0.360	−0.027	0.220	0.466 *	0.774 **

C = control group; LS = lifestyle intervention group; BS = bariatric surgery group; BradyG = bradygasria; PR = preprandial; NormoG = normogastria; TachyG = tachygastria. Min10 = the first 10 min in postprandial recording; Min20 = the period between 10 to 20 min in postprandial recording; Min30 = the period between 20 to 30 min in postprandial recording; ADF = average dominant frequency; * correlation is significant at the 0.05 level (2-tailed); ** correlation is significant at the 0.01 level (2-tailed).

## Data Availability

The datasets for this study are part of an ongoing project. However, requests for data may be made to any qualified researcher.

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
