# Peer review of "Gastric Myoelectric Activity and Body Composition Changes during Weight Loss via Sleeve Gastrectomy and Lifestyle Modification: Prospective Cohort Study"

_healthcare, 2023, doi:10.3390/healthcare11081105_

Round 1

Reviewer 1 Report

The authors for the first time showed improved body composition parameters in patients who undertook sleeve gastrectomy more than patients who went through lifestyle intervention. Remarkably, the current study revealed that fat mass change is significantly associated with Gastric Myoelectric Activity changes in both bariatric surgery and lifestyle intervention. Interestingly in the LS group,  amount of fat mass loss was negatively correlated with bradygastria times and positively with pre-prandial and early post-prandial average dominant frequency. Although there is superior weight-loss in BS patients , LS produced modest normalization of the GMA while preserving fat-free mass. Hence, the study supports normalizing GMA disturbances as a possible mechanism of weight loss after BS. The authors did a good job in discussing the results in a subtle way.

Minor comments

Is it bradygasteria or bradygastria? Please correct it or explain? Add references if necessary

Author Response

Response to Reviewer 1

Manuscript ID:  healthcare-2252545

Manuscript Title: “Gastric myoelectric activity and body composition changes during weight loss via bariatric surgery and lifestyle modification: Prospective cohort study”

We thank the reviewers for their careful examination of the manuscript and appreciate the useful suggestions to improve the quality of our paper. Our point-by-point response to the reviewers' comments is given below. Changes in the manuscript are indicated in red font. Please note that the pages and line numbers mentioned in the reviewers’ comments refer to the original manuscript, whereas those in the authors’ reply refer to the revised manuscript.

Comments from the Editors and Reviewers:

  1. The authors for the first time showed improved body composition parameters in patients who undertook sleeve gastrectomy more than patients who went through lifestyle intervention. Remarkably, the current study revealed that fat mass change is significantly associated with Gastric Myoelectric Activity changes in both bariatric surgery and lifestyle intervention. Interestingly in the LS group,  amount of fat mass loss was negatively correlated with bradygastria times and positively with pre-prandial and early post-prandial average dominant frequency. Although there is superior weight-loss in BS patients , LS produced modest normalization of the GMA while preserving fat-free mass. Hence, the study supports normalizing GMA disturbances as a possible mechanism of weight loss after BS. The authors did a good job in discussing the results in a subtle way.

Answer: Thank you for pointing this out.

  1. Minor comments

Is it bradygasteria or bradygastria? Please correct it or explain. Add references if necessary.

Answer: Sorry for this oversight and typo mistake. It is bradygastria and was corrected throughout the manuscript.

Reviewer 2 Report

I am pleased to read this interesting paper on gastric myoelectric activity and body composition changes after various types of interventions. A comparison was made between 3 groups: after sleeve gastrectomy, after lifestyle intervention and a control group.

I have some questions: 

- My main question is: why did the authors choose this approach to measure gastrid motility? Do you estimate this technique superior to for example gastric emptying studies using scintigraphy? What are the disadvantages of measuring GMA? Is it reliable after gastric surgery in your opinion? 

- Were there lifestyle interventions in the BS group (dietary advice, physiotherapy, etc.)? 

- Please improve the sentence in line 49, cosmetic changes are not the reason to perform metabolic surgery

- Could you explain how bradygastria - normogastria - tachygastria are defined? is this age/sex/... dependent? 

- Could you explain what is meant with 'average dominant frequency' How is it defined, what is the clinical significance of this measurement? 

- could you explain the randomization process, why did group LS have so little participants to start with? 

- were participants in the LS/C group offered metabolic surgery after the study period? if so, it would be interesting to compare GMA measurements in the same subject!

- i would propose using time-periods (eg. 3 month-6 month), instead of 'final', 'short-term', 'long-term', 'mid', etc. This is easier to follow/read to me. 

- sentence 192-195: The alinea starts with the BS group, however than the authors write 'compared to baseline assesment in the LS group; .--> This is a bit confusing, if i understand correctly, this sentence is refering to the LS group? 

- in the tables: BL-bradayG(%) etc.: it is unclear to me how i should interprete; are the numbers always a percentage of baseline value? 

- the authors make statements on pre- and post-prandial status. I am a bit confused: do participants receive food in the experiment? I am not convinced that water-loading can be considered equal to a fed state? Can you elaborate on this? It would be interesting to compare water versus food intake.. 

- I think the discussion could be a bit more focused: why are these findings important? Where do they correspond to what is already known, where do they not?  Do the authors see options for further research? 

Author Response

Response to Reviewer 2

Manuscript ID:  healthcare-2252545

Manuscript Title: “Gastric myoelectric activity and body composition changes during weight loss via bariatric surgery and lifestyle modification: Prospective cohort study”

We thank the reviewers for their careful examination of the manuscript and appreciate the useful suggestions to improve the quality of our paper. Our point-by-point response to the reviewers' comments is given below. Changes in the manuscript are indicated in red font. Please note that the pages and line numbers mentioned in the reviewers’ comments refer to the original manuscript, whereas those in the authors’ reply refer to the revised manuscript.

Comments from the Editors and Reviewers:

  1. I am pleased to read this interesting paper on gastric myoelectric activity and body composition changes after various types of interventions. A comparison was made between 3 groups: after sleeve gastrectomy, after lifestyle intervention and a control group.

Answer: Thank you for pointing this out.

  1. I have some questions: 

- My main question is: why did the authors choose this approach to measure gastric motility? Do you estimate this technique superior to for example gastric emptying studies using scintigraphy? What are the disadvantages of measuring GMA? Is it reliable after gastric surgery in your opinion? 

Answer: Thanks for this good question. The mechanical contractility of the stomach or any contractile organ is preceded by electrical activity that originates from specialized pacemaker cells. The firing frequency and amplitude of this electric activity determine the motility and emptying of the stomach. EGG measures this electrical activity while scintigraphy measures gastric emptying which is mechanical motility. A combination of both techniques is for sure more comprehensive. the EGG and gastric emptying scintigraphy were found to be complimentary for the assessment of dyspepsia. Delayed gastric emptying was able to be predicted using the EGG with a specificity of 80% and a sensitivity of 55-60% (Chen JDZ, Lin L, Pan J, McCallum RW. Abnormal gastric myoelectrical activity and delayed gastric emptying in patients with symptoms suggestive of gastroparesis. Dig Dis Sci. 1996;41:1538–1545). However, this work focused on gastric electric activity which determines the potential for starting motility and explains many functional disorders in the stomach. We assume that obesity produces functional disturbances of the GMA which will be ameliorated with weight loss. EGG is noninvasive, clinically valid, and tolerated by most patients.  In gastric surgeries including sleeve gastrectomy, some of the pacemaker cells (ICC) are removed. However, EGG may be applicated for evaluating motility and autonomic functions of the remnant stomach (Kim HY, Park SJ, Kim YH. Clinical application of electrogastrography in patients with stomach cancer who undergo distal gastrectomy. J Gastric Cancer. 2014 Mar;14(1):47-53. doi: 10.5230/jgc.2014.14.1.47. Epub 2014 Mar 31. PMID: 24765537; PMCID: PMC3996249.)

  1. Were there lifestyle interventions in the BS group (dietary advice, physiotherapy, etc.)? 

Answer: Apart from the traditional dietary management of bariatric surgery patients, participants of the BS group didn’t instruct to do specific lifestyle interventions.

  1. - Please improve the sentence in line 49, cosmetic changes are not the reason to perform metabolic surgery.

Answer: thanks for this point. It changed to “weight changes”

  1. - Could you explain how bradygastria - normogastria - tachygastria are defined? is this age/sex/... dependent? 

Answer: Definitions were mentioned in lines 129-131 and in the footnote of Figure 2. The definitions are not age/sex-dependent. However, the dominant frequency itself may be affected by age, gender, and phase of the menstrual cycle (Parkman HP, Harris AD, Miller MA, Fisher RS. Influence of age, gender, and menstrual cycle on the normal electrogastrogram. Am J Gastroenterol. 1996 Jan;91(1):127-33. PMID: 8561112). in our sample all were adults and male-female ratio was similar in all groups (Females represented 46% of the C group, 50% of the LS group, and 44% of the BS group). Another opinion stated that age, gender, and obesity did not appear to affect the EGG parameters of mean dominant frequency or the power ratio (Riezzo G, Russo F, Indrio F. Electrogastrography in adults and children: the strength, pitfalls, and clinical significance of the cutaneous recording of the gastric electrical activity. Biomed Res Int. 2013;2013:282757. doi: 10.1155/2013/282757. Epub 2013 May 25. PMID: 23762836; PMCID: PMC3677658.)

  1. - Could you explain what is meant with 'average dominant frequency' How is it defined, what is the clinical significance of this measurement? 

Answer: thanks for this question. The definition was added to lines 133 to 136. EGG recording is a combination of the gastric signal (the main component) and noise from other excitable tissue. The average dominant frequency is the dominant frequency, which is believed to be of gastric origin and at which the power in the power spectrum has a peak value. The average dominant frequency of the EGG accurately reflects the frequency of the gastric slow wave. The normal range of the dominant frequency of the EGG is between 2 to 4 cpm.  (Chen JD. Spectral analysis of electrogastrogram and its clinical significance. World J Gastroenterol 1996; 2(Suppl1): 9-11 [DOI: 10.3748/wjg.v2.iSuppl1.9)

  1. - could you explain the randomization process, why did group LS have so little participants to start with?

Answer: Unfortunately we didn’t do randomization. the study was a nonrandomized clinical trial. The randomization word is now deleted from Figure 1. In our locality, patients with obesity prefer surgical solutions and compliance with LS is relatively low. This may explain why LS have so little participants.

  1. - were participants in the LS/C group offered metabolic surgery after the study period? if so, it would be interesting to compare GMA measurements in the same subject!

Answer: We did not follow the patients after the end of the study. It is not allowed to track their files without IRB permission for a new study.

  1. - i would propose using time-periods (eg. 3 month-6 month), instead of 'final', 'short-term', 'long-term', 'mid', etc. This is easier to follow/read to me. 

Answer: Thanks for the suggestion. Done

  1. - sentence 192-195: The alinea starts with the BS group, however than the authors write 'compared to baseline assesment in the LS group; .--> This is a bit confusing, if i understand correctly, this sentence is refering to the LS group? 

Answer: Sorry for this oversight. It is now clear.

  1. - in the tables: BL-bradayG(%) etc.: it is unclear to me how i should interprete; are the numbers always a percentage of baseline value? 

Answer: In the first column the units were indicated between brackets i.e. (%) = percentage, cpm= cycle per min, ..etc). In the table footnote, it is indicated that PR = Preprandial, BradyG = Bradygatria, …etc.    

  1. - the authors make statements on pre- and post-prandial status. I am a bit confused: do participants receive food in the experiment? I am not convinced that water-loading can be considered equal to a fed state? Can you elaborate on this? It would be interesting to compare water versus food intake...

Answer: In the EGG procedure we consider recording before loading with any stimulus (water or food) as a pre-prandial and recording after as a postprandial. A comparison of water vs caloric fluids or food was done before but this is beyond the scope of this work.

https://doi.org/10.1080/003655202760373344

https://doi.org/10.1111/nmo.14376

  1. - I think the discussion could be a bit more focused: why are these findings important? Where do they correspond to what is already known, and where do they not?  Do the authors see options for further research? 

Answer: We did our best to focus and conclude our main finding at the discussion (paragraph from line 324 to line 337)

Reviewer 3 Report

Dear authors,

Congratulations with such a well-planned and carefully implemented study. My comments are mild. Please, consider the following:

1. Consider changing the manuscript title. In fact, you only had sleeve gastrectomy patients. Therefore, reflect this in the title. Bariatric surgery is too broad, avoid using this term in the title.

2. Please, be careful with the term "randomized" that you use in Figure 1. This was not an actual randomization, but distribution of patients between intervention groups.

3. Line 110. The sentence "Where all study measurements were done. " appears to be incomplete.

4. For better visualization, avoid using acronyms in the legend of Figure 3 ("C", "LS", "BS"). Please, consider using the full terms instead.

5. Although your study has obvious strong sides, it has certain limitations too. Please, discuss them. One of the significant limitations that you have is uneven drop-out of individuals from different study groups.

Author Response

Response to Reviewer 3

Manuscript ID:  healthcare-2252545

Manuscript Title: “Gastric myoelectric activity and body composition changes during weight loss via bariatric surgery and lifestyle modification: Prospective cohort study”

We thank the reviewers for their careful examination of the manuscript and appreciate the useful suggestions to improve the quality of our paper. Our point-by-point response to the reviewers' comments is given below. Changes in the manuscript are indicated in red font. Please note that the pages and line numbers mentioned in the reviewers’ comments refer to the original manuscript, whereas those in the authors’ reply refer to the revised manuscript.

Comments from the Editors and Reviewers:

  1. Congratulations with such a well-planned and carefully implemented study. My comments are mild. Please, consider the following:.

Answer: Thank you for your kind words.

  1. Consider changing the manuscript title. In fact, you only had sleeve gastrectomy patients. Therefore, reflect this in the title. Bariatric surgery is too broad, avoid using this term in the title.

Answer: Thanks for this suggestion. The title was changed.

  1. Please, be careful with the term "randomized" that you use in Figure 1. This was not actual randomization, but a distribution of patients between intervention groups.

Answer: Changed. Thanks for the comment.

  1. Line 110. The sentence "Where all study measurements were done. " appears to be incomplete.

Answer: Sorry for this oversight. It is now corrected.

  1. For better visualization, avoid using acronyms in the legend of Figure 3 ("C", "LS", "BS"). Please, consider using the full terms instead.

Answer: Thanks for this suggestion. It is updated.

  1. Although your study has obvious strong sides, it has certain limitations too. Please, discuss them. One of the significant limitations that you have is uneven drop-out of individuals from different study groups.

Answer: Thanks for this comment. Limitations were added (lines 316-320).
